# Steering LLM Thinking with Budget Guidance

## Abstract

Recent deep-thinking large language models often reason extensively to improve performance, but such lengthy reasoning is not always desirable, as it incurs excessive inference costs with disproportionate performance gains. Controlling reasoning length without sacrificing performance is therefore important, but remains challenging, especially under tight thinking budgets. We propose *budget guidance*, a simple yet effective method for steering the reasoning process of LLMs toward a target budget without requiring any LLM fine-tuning. Our approach introduces a lightweight predictor that models a Gamma distribution over the remaining thinking length during next-token generation. This signal is then used to guide generation in a soft, token-level manner, ensuring that the overall reasoning trace adheres to the specified thinking budget. *Budget guidance* enables natural control of the thinking length, along with significant token efficiency improvements over baseline methods on challenging math benchmarks. For instance, it achieves up to a 26% accuracy gain on the MATH-500 benchmark under tight budgets compared to baseline methods, while maintaining competitive accuracy with only 63% of the thinking tokens used by the full-thinking model. *Budget guidance* also generalizes to broader task domains and exhibits emergent capabilities, such as estimating question difficulty.

## 1 Introduction

With the recent success of deep-thinking large language models (LLMs) – such as OpenAI O1 (Jaech et al., 2024), DeepSeek R1 (Guo et al., 2025), and Qwen3 (Yang et al., 2024a;b), which are capable of generating long sequences of thoughts to achieve better performance – there has been a growing need to control the reasoning length of these models while maintaining the performance, because many deep-thinking LLMs often incur excessive inference costs with disproportionate performance gain. For example, in Figure 1, we show a response from a deep-thinking model that, while correct, is unnecessarily long. Such extensive reasoning is not always desirable, and there are cases where we need to impose a budget to limit the extent of reasoning, particularly in scenarios that demand real-time interaction, such as customer-facing chatbots, where excessive latency can degrade user experience and responsiveness.

Existing thinking budget control methods can be roughly divided into two categories with complementary strengths. The first category is fine-tuning methods, which fine-tune deep-thinking LLMs on specially curated dataset (Han et al., 2024) or with budget-aware reward to enable budget control capabilities (Hou et al., 2025). Fine-tuning methods have been shown effective in changing the reasoning length while keeping competitive performance because they allow LLMs to fundamentally restructure and optimize their reasoning behavior according to the given budget. However, they come with two main drawbacks. First, fine-tuning an LLM is costly, requiring substantial computational resources and time. Second, directly fine-tuning the LLM may potentially alter its behavior in unexpected ways, such as compromising safety (Qi et al., 2023).

The second category of methods is the inference-time methods (Ma et al., 2025; Muennighoff et al., 2025), which seek to alter the reasoning behavior at inference time. While these approaches do not involve fine-tuning, they often result in sub-optimal reasoning behaviors and significant performance degradation, because the intervention at inference time are often heuristic and overly simple, breaking the integrity of the original reasoning process. For example, one well-known inference-time method is *budget forcing* (Muennighoff et al., 2025) which terminates the model's reasoning as soon as the thinking budget is reached, as described in Figure 1. While this method offers strict control over the

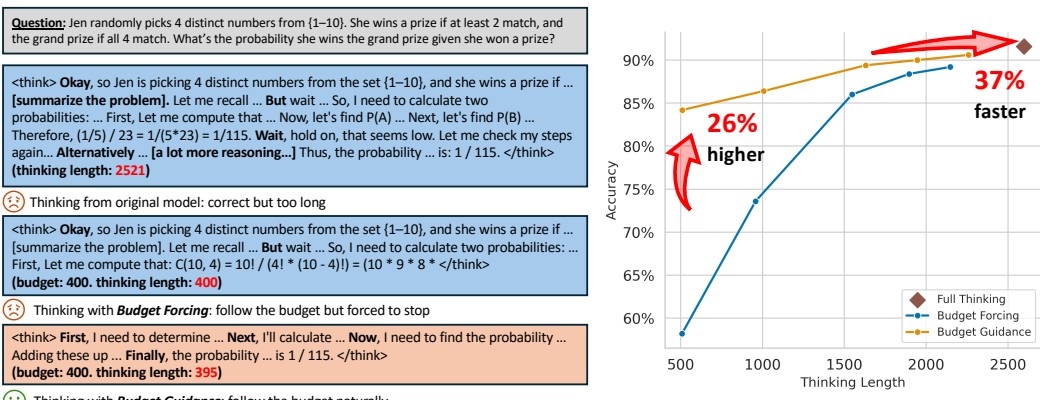

Figure 1: Deep-thinking models often produce excessively long reasoning traces, leading to high latency and unnecessary computation. Existing inference-time methods like *budget forcing* rely on simplistic heuristics such as abruptly stopping, which can result in incomplete reasoning and degraded answer quality. In contrast, our method, *budget guidance*, steers the reasoning process toward the target budget in a smoother and more natural way, without any LLM fine-tuning.

number of generated tokens, abruptly interrupting the model may cut off unfinished thoughts and force premature answers, often leading to incorrect outputs.

In short, an important bottleneck in the task of thinking budget control lies in the tradeoff between *non-intrusiveness* (in inference-time approaches) and *optimality of the reasoning chain* (in fine-tuning approaches). This leads to our central research question: *Can we design a flexible inference-time budget control approach (without fine-tuning) that still allows for wholistic, principled restructuring of the reasoning process to maintain its quality under budget?*

In this paper, we introduce *budget guidance*, a novel approach that employs a lightweight auxiliary module to enable test-time control over the reasoning length of LLMs. Inspired by the principle of classifier guidance in diffusion models (Dhariwal & Nichol, 2021), we train an auxiliary predictor that predicts the probability distribution of the remaining reasoning length at each reasoning step. The predicted length distribution is then used to modulate the LLM generation probability, effectively turning it into a budget-conditional generation probability. Our method avoids the direct fine-tuning of LLMs, while providing flexible and accurate control over the reasoning process. It can be seamlessly integrated into existing inference pipelines, and adapts to a wide range of models, thinking budgets, and tasks.

Our experiments have revealed several key highlights of our method. First, *budget guidance* exhibits a remarkable trade-off between thinking length and performance. For example, as shown in Figure 1, on MATH-500 benchmark (Hendrycks et al., 2021) *budget guidance* can reduce the full thinking length by 37% with minimal accuracy degradation, while being 26% higher in accuracy than *budget forcing* baseline under tight budget. Second, the auxiliary predictor is very successful in predicting the thinking length, effectively considering task difficulty and instruction type. Thus, it can accurately guide the thinking process under various budgets. Finally, our method demonstrates surprising generalizability across domains – an auxiliary predictor trained on one dataset can also work well in other datasets and domains.

We summarize our contributions as follows:

- We propose *budget guidance*, a novel test-time method for steering the reasoning process of LLMs toward a specified thinking budget, without requiring any fine-tuning of the LLM itself.
- We design a lightweight predictor that models a Gamma distribution over the remaining reasoning length based on the current generation context, and uses this signal to guide LLM generation toward a target thinking budget.
- *Budget guidance* achieves strong trade-offs between thinking length and accuracy across multiple benchmarks, and demonstrates cross-domain generalization, enabling effective budget control and accurate thinking length prediction.

## 2 RELATED WORKS

### 2.1 EFFICIENT LLM REASONING

Efficiency in large language model (LLM) reasoning has been studied through two main paradigms: *fine-tuning based methods* and *inference-time steering*. Fine-tuning methods such as ThinkPrune (Hou et al., 2025), Z1 (Yu et al., 2025), and COCONUT (Hao et al., 2024) shorten reasoning traces via reinforcement learning, curriculum-style training on variable-length data, or continuous latent representations. While effective, these methods typically rely on expensive LLM fine-tuning and primarily aim to *reduce* the length of reasoning, rather than to *control* it. More recent approaches (Han et al., 2024; Muennighoff et al., 2025) have begun exploring methods to control the reasoning length, either through heuristic rules or model fine-tuning. In contrast, we propose a simple yet effective alternative: a fine-tuning-free approach that naturally steers the reasoning process to adhere to a specified thinking budget, enabling more efficient and flexible inference.

Inference-time steering, in contrast, intervenes directly during decoding without fine-tuning the LLM model. Dynasor (Fu et al., 2025) and DEER (Yang et al., 2025b) dynamically allocates compute by probing intermediate steps and early terminating confident cases. In contrast, SEAL (Chen et al., 2025) calibrates reasoning traces by applying lightweight latent-space interventions to suppress reflection and transition thoughts, thereby reducing redundancy during inference. While effective, these methods primarily optimize efficiency heuristically and do not offer fine-grained control over reasoning length. Simpler strategies include NoThinking (Ma et al., 2025), which bypasses reasoning altogether but typically suffers from severe accuracy loss. The most widely adopted approach that enables explicit steering is budget forcing (Muennighoff et al., 2025), used in real-world applications such as Claude 3.7 Sonnet[1] and Qwen3 Yang et al. (2025a). It enforces a hard token cutoff to guarantee that reasoning length stays within a given budget. Although surprisingly effective in practice, this method leaves reasoning patterns untouched and forcibly terminates the reasoning once the budget is reached. In contrast, our approach offers smooth and fine-grained control over reasoning length, eliminating the need for heuristic rules or hard cutoffs.

### 2.2 GUIDANCE AND GUIDED GENERATION

The term *guidance* originates primarily from the diffusion model literature, where it denotes the ability to steer the generative process, often through truncated or low-temperature sampling, by reducing the variance or range of noise inputs to the generative model at sampling time (Ho & Salimans, 2022). This effectively transforms an unconditional diffusion model into a conditional one, enabling it to generate targeted outputs. One of the earliest examples is *classifier guidance* (Dhariwal & Nichol, 2021), which modifies the diffusion score by incorporating the gradient of the log-likelihood from an auxiliary classifier, thereby biasing the sampling process toward desired content. This can be viewed as a form of guided generation, where image generation is conditioned on the output of a classifier.

A similar notion of guided generation has emerged in the context of LLMs, where it typically refers to constraining the model's output to satisfy structural requirements, such as context-free grammars, to ensure syntactic correctness for downstream applications (Willard & Louf, 2023).

To the best of our knowledge, our work is the first to extend the idea of guided generation to a new dimension: *budget-conditioned generation*. Specifically, we introduce a novel form of guidance that softly steers the LLM's generation to meet a specified thinking budget, enabling efficient and controlled reasoning without compromising output quality.

## 3 BUDGET GUIDANCE

We now introduce our method in detail. In Section 3.1, we begin by formulating the budget-conditioned generation problem and present the overall *budget guidance* framework, which draws inspiration from classifier guidance (Dhariwal & Nichol, 2021) in diffusion models. Section 3.2 describes the design of our proposed auxiliary thinking length predictor, which estimates the distribution over remaining reasoning length at each decoding step. In Section 3.3, we outline the training procedure for the predictor using reasoning traces. Section 3.4 introduces the model architecture

---

[1] https://www.anthropic.com/news/visible-extended-thinking

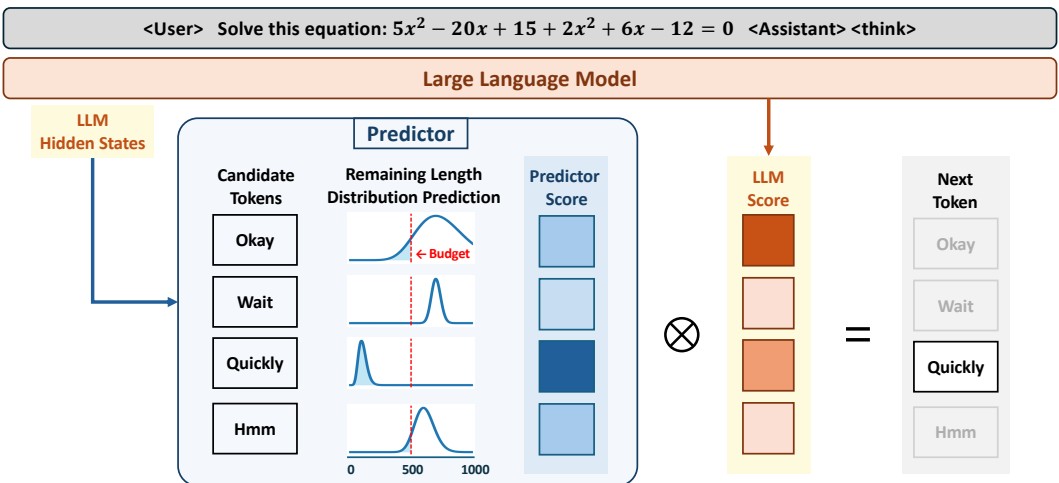

Figure 2: An overview of *budget guidance*. A lightweight predictor uses the LLM's hidden states to predict a Gamma distribution over the remaining reasoning length for each candidate token. We then use the CDF of Gamma distribution to compute a predictor score, which is combined with the LLM's output score to guide generation. The result is soft, token-level steering toward budget-conditioned reasoning without any LLM fine-tuning.

of the predictor, which is designed to be lightweight and inference-efficient. Finally, Section 3.5 presents a simple modulation-skipping strategy to further reduce computational overhead during decoding. An illustration of our method is provided in Figure 2.

## 3.1 THE BUDGET GUIDANCE FRAMEWORK

The overall framework of our method follows the classifier guidance framework in diffusion generation (Dhariwal & Nichol, 2021), thus we name our framework *budget guidance*. Specifically, denote $X$ as the input question, $Y_{<t}$ as the LLM's output thinking process up to token $t$, and $Y_t$ as the LLM's output at token $t$. The LLM generation process essentially involves sampling from the following *budget-unconditional distribution*, $p(Y_t|X, Y_{<t})$.

However, when there is a budget constraint, we would need to draw from a *budget-conditional distribution*. Formally, denote $L_t$ as the random variable indicating the *remaining length* of the thinking process from token $t$. For example, if the overall thinking length is $l$ (i.e., the `</think>` token occurs at token $l$), then $L_t = l - t$. Given the thinking budget limit $\bar{l}$, the budget-conditional distribution is defined as $p(Y_t|X, Y_{<t}, L_t \leq \bar{l} - t)$.

According to Bayes' rule, the budget-conditional distribution can be computed from the budget-unconditional distribution as follows

$$\underbrace{p(Y_t|X, Y_{<t}, L_t \leq \bar{l} - t)}_{\text{budget-conditional}} \propto \underbrace{p(Y_t|X, Y_{<t})}_{\text{budget-unconditional}} \cdot Pr(L_t \leq \bar{l} - t|X, Y_{<t}, Y_t). \quad (1)$$

Therefore, at each token $t$, generating from the budget-conditional distribution involves three steps. First, compute the unconditional distribution, which is simply performing a forward pass of the LLM. Second, predict the remaining length distribution, $Pr(L_t \leq \bar{l} - t|X, Y_{<t}, Y_t)$. Finally, use the remaining length distribution to modulate the unconditional distribution and then renormalize.

Therefore, within budget guidance framework, our task boils down to computing $Pr(L_t \leq \bar{l} - t|X, Y_{<t}, Y_t)$. To this end, we introduce a lightweight auxiliary thinking length predictor, which we describe in detail over the next three subsections.

## 3.2 AN AUXILIARY THINKING LENGTH PREDICTOR

Denote the LLM vocabulary size as $n$, and denote the vocabulary as $\mathcal{V} = \{v_1, \ldots, v_n\}$. At each token $t$, the LLM outputs an $n$-dimensional unconditional probability vector (which we denote as

$\boldsymbol{u}_t$):
$$\boldsymbol{u}_t = [p(Y_t = v_1|X, Y_{<t}), \ldots, p(Y_t = v_n|X, Y_{<t})]. \tag{2}$$

According to Equation 1, the predictor needs to predict an $n$-dimensional vector (which we denote as $\boldsymbol{a}_t$):
$$\boldsymbol{a}_t = [Pr(L_t \le \bar{l} - t|X, Y_{<t}, Y_t = v_1), \cdots, Pr(L_t \le \bar{l} - t|X, Y_{<t}, Y_t = v_n)], \tag{3}$$

so that the budget-conditional probability vector, which we denote as $\boldsymbol{c}_t$, can be computed by element-wise multiplying the two vectors and renormalize:
$$\boldsymbol{c}_t = normalize(\boldsymbol{u}_t \circ \boldsymbol{a}_t). \tag{4}$$

Equation 3 indicates that the predictor needs to accomplish a rather intensive task: At each token $t$, given the question $X$ and all the context generated so far $Y_{<t}$, the auxiliary predictor needs to ❶ traverse all possible values for $Y_t$ across the vocabulary, ❷ for each possible value, predict what would be the remaining length if $Y_t$ took on this value (that is $n$ probability distributions in total), and ❸ compute the cumulative probability up to $\bar{l} - t$ for each distribution.

To simplify the task, we parameterize each predicted distribution as a Gamma distribution for $\log(L_t)$:
$$p(L_t|X, Y_{<t}, Y_t = v_i) = Gamma(\log(L_t); \lambda_t(v_i), \alpha_t(v_i)), \tag{5}$$

where $Gamma(\cdot; \lambda, \alpha)$ represents the probability density function (PDF) of the Gamma distribution, with the shape parameter $\lambda$ and rate parameter $\alpha$. We model the distribution over $\log(L_t)$ instead of $L_t$ directly to better capture the dynamic range of thinking lengths.

With the Gamma distribution assumption, instead of predicting $n$ probability distributions, we only needs to predict two $n$-dimensional vectors: $\boldsymbol{\lambda}_t = [\lambda_t(v_1), \ldots, \lambda_t(v_n)]$ and $\boldsymbol{\alpha}_t = [\alpha_t(v_1), \ldots, \alpha_t(v_n)]$. The cumulative probability, $\boldsymbol{a}_t$, can be computed from the predicted $\boldsymbol{\lambda}_t$ and $\boldsymbol{\alpha}_t$ by the known closed-form cumulative distribution function (CDF) of the Gamma distribution.

### 3.3 Training the Predictor

To train the predictor, we need to collect a dataset of reasoning chains produced by the target LLM. Formally, the data in the dataset takes the following form: $\mathcal{D} = \{(x, y_{1:l}, l)\}$, where $x$ is the input question, $y_{1:l}$ is the LLM-generated reasoning chain, and $l$ is the length of the reasoning chain. Note that the task dataset from which reasoning chain length training data are generated is not the same as the inference dataset (not even the same task), as we will show that the trained predictor has good dataset and task generalizability. For simplicity, in our training, we focus on math reasoning and use the OpenR1-Math-220k dataset (Face, 2025).

For each training datum, $(x, y_{1:l}, l)$, we feed the information of a partial reasoning chain to the predictor, truncated at different positions, and train the predictor to predict the remaining length. We adopt the maximum log likelihood objective for the gradient descent training. Formally, denote the parameters of the auxiliary predictor as $\boldsymbol{\theta}$. Then the training objective can be written as

$$\max_{\boldsymbol{\theta}} \mathbb{E}_{(x,y_{1:l},l)\sim\mathcal{D}} \left[ \sum_{t=1}^{l-1} \log \left( p_{\boldsymbol{\theta}}(L_t = l - t|X = x, Y_{<t} = y_{<t}, Y_t = y_t) \right) \right], \tag{6}$$

where $p_{\boldsymbol{\theta}}(\cdot)$ represents the predicted PDF by the auxiliary predictor, as shown in Equation 5.

### 3.4 Architecture of the Predictor

The predictor is designed to be lightweight enough to avoid significant computational overhead during decoding, yet expressive enough to capture both the input question and the ongoing reasoning context to produce a meaningful estimate of the remaining reasoning length. To this end, we adopt BERT-base (Devlin et al., 2019) as the backbone of our predictor. Its input consists of the concatenated hidden states from all layers of the last generated token of the target LLM, which encode rich semantic information about both the input question and the reasoning history. A linear projection maps the LLM's hidden dimensionality to the predictor's input space, and a [CLS] token is used to summarize the hidden states. The final [CLS] representation is passed through another linear projection to produce an output matrix $M \in \mathbb{R}^{n \times 2}$, where each row corresponds to the parameters $\boldsymbol{\lambda}_t$ and $\boldsymbol{\alpha}_t$ of a Gamma distribution. A softplus activation (Dugas et al., 2000) is applied to ensure both parameters are non-negative.

### 3.5 SKIPPING MODULATION

Ideally, probability modulation in Equation 4 would be applied at every decoding step $t$. To reduce computational overhead, however, we apply it only at the start of each reasoning paragraph, indicated by newline delimiters, where uncertainty is typically highest. The modulation is thus defined as:

$$c_t = \begin{cases} \text{normalize}(\boldsymbol{u}_t \circ \boldsymbol{a}_t), & \text{if } t \text{ is the start of a reasoning paragraph} \\ \boldsymbol{u}_t, & \text{otherwise} \end{cases} \tag{7}$$

Empirically, we find that our predictor leads to negligible latency overhead, which is discussed in detail in Appendix A.

## 4 EXPERIMENTS

### 4.1 SETTINGS

**Training.** We apply our method to three deep-thinking models: *DeepSeek-R1-Distill-Qwen-7B* (R1-7B) (Guo et al., 2025), *DeepSeek-R1-Distill-Qwen-32B* (R1-32B) (Guo et al., 2025), and *Qwen3-8B* (Yang et al., 2024a;b). Training is conducted on *OpenR1-Math-220k* (Face, 2025), a dataset of 220k math problems from *NuminaMath 1.5* (Li et al., 2024) with reasoning traces generated by DeepSeek R1. We apply a simple data augmentation technique (detailed in the Appendix) to double the dataset size. During training, the LLMs are frozen and only the predictor is updated. We train for one epoch using a batch size of 8 and a constant learning rate of $1.0 \times 10^{-4}$ after warmup. Training takes 15 hours for R1-7B and Qwen3-8B, and 35 hours for R1-32B, using 8 NVIDIA H100 GPUs. All evaluations are conducted on the same hardware setup.

**Evaluation.** We evaluate our method on four representative math reasoning benchmarks: **MATH-500** (Hendrycks et al., 2021), **AIME-2024** (Art of Problem Solving, n.d.a), **AMC** (Art of Problem Solving, n.d.b) (including both AMC12 2022 and AMC12 2023), and the math subset from **Olympiad-Bench** (He et al., 2024). These benchmarks cover diverse mathematical topics, including arithmetic, algebra, combinatorics, *etc.*, and span a broad range of difficulty levels.

Besides math benchmarks, we also extend our evaluation to broader domains to test the out-of-domain transferability of our math-data-trained predictor. Specifically, we further evaluate on **GPQA Diamond** (Rein et al., 2024) for scientific reasoning, **FOLIO** (Han et al., 2022) for logical reasoning, the numerical reasoning subset from **TableBench** (Wu et al., 2025) for tabular numerical reasoning, and **LiveCodeBench** (Jain et al., 2024) (following (Guo et al., 2025)) for code reasoning.

All experiments are conducted in a zero-shot manner, *i.e.*, we do not perform further fine-tuning on the training sets of the evaluation benchmarks. We use greedy decoding for all evaluation.

**Baselines.** We compare our method with other methods that also do not finetune the LLM. Our main baseline is *budget forcing* (Muennighoff et al., 2025), which enforces a hard token limit by appending an end-of-thinking delimiter (and optionally "Final Answer:") to trigger early exit and force the model to produce its best guess. We use their open-sourced codebase for evaluation. We also include *NoThinking* (Ma et al., 2025) as a baseline, which bypasses the reasoning stage by inserting a fixed phrase as the thinking process: `Okay, I think I have finished thinking`. We also report results from the original model, which serves as a reference for full thinking. For the accuracy-thinking length trade off experiments in Section 4.2.2, we additionally compare against Dynasor (Fu et al., 2025), which is a recent inference time steering method that uses a strategy called *Probe-In-The-Middle* that can early stop confident reasoning traces. It does not require LLM fine-tuning. We adopt the default token-saving level (mid) provided in their official codebase.

### 4.2 MAIN RESULTS

#### 4.2.1 EVALUATION ON MATH REASONING BENCHMARKS

Since the predictor is trained on math data, we first evaluate its performance on math reasoning benchmarks to assess in-domain effectiveness. We set the thinking budget to approximately half the original model's full thinking length and ensure the average thinking length (denoted as #Tokens) comparable between our method and the baseline, and report the task accuracy.

Table 1: Evaluation results on math benchmarks.

| | MATH-500 | | AIME-2024 | | AMC | | OlympiadBench | |
|---|---|---|---|---|---|---|---|---|
| | Acc. | #Tokens | Acc. | #Tokens | Acc. | #Tokens | Acc. | #Tokens |
| *DeepSeek-R1-Distill-Qwen-7B* | | | | | | | | |
| Thinking | 91.6 | 2598 | 36.7 | 4446 | 78.3 | 4338 | 56.9 | 3960 |
| NoThinking | 74.8 | - | 23.3 | - | 47.0 | - | 40.4 | - |
| Budget Forcing | 86.0 | 1547 | 16.7 | **2015** | 55.4 | 1872 | 47.1 | 1844 |
| **Budget Guidance** | **88.2** | **1329** | **33.3** | 2046 | **60.2** | **1768** | **54.2** | **1755** |
| *DeepSeek-R1-Distill-Qwen-32B* | | | | | | | | |
| Thinking | 93.2 | 2226 | 70.0 | 7694 | 77.1 | 4156 | 61.9 | 3435 |
| NoThinking | 68.2 | - | 20.0 | - | 47.0 | - | 41.9 | - |
| Budget Forcing | 86.4 | 1525 | 40.0 | 2936 | 50.6 | 1567 | 50.7 | **1797** |
| **Budget Guidance** | **90.0** | **1288** | **56.7** | 2873 | **69.9** | **1528** | **57.8** | 1820 |
| *Qwen3-8B* | | | | | | | | |
| Thinking | 96.2 | 4613 | 73.3 | 13660 | 91.6 | 8740 | 71.7 | 9424 |
| NoThinking | 84.8 | - | 33.3 | - | 56.6 | - | 53.5 | - |
| Budget Forcing | 90.2 | 2545 | 43.3 | 4010 | 77.1 | **3807** | 61.6 | 3712 |
| **Budget Guidance** | **93.0** | **2062** | **50.0** | **3981** | **80.7** | 3869 | **65.6** | **3639** |

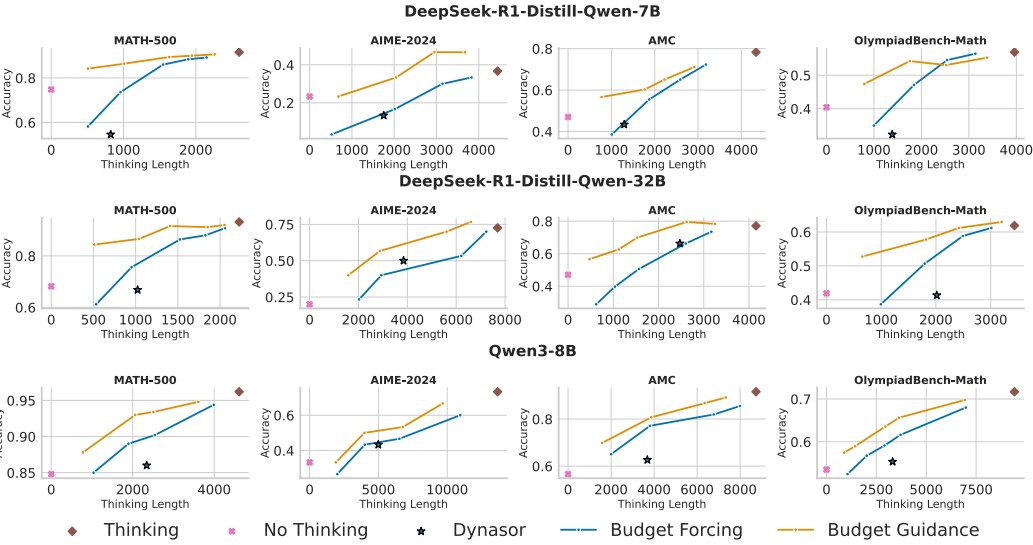

Figure 3: Accuracy vs. thinking length on math benchmarks.

Table 1 summarizes the evaluation results on math reasoning benchmarks. Across all three models and four datasets, *budget guidance* consistently outperforms *budget forcing* under comparable average thinking lengths, effectively reducing the reasoning length without causing significant accuracy degradation. Compared to *NoThinking*, *budget guidance* achieves substantially higher performance, indicating that the reasoning traces are non-trivial and contribute meaningfully to task success.

These improvements are consistent across different model sizes (7B to 32B) and model families (DeepSeek vs. Qwen3), highlighting the general applicability of our approach to diverse deep-thinking LLMs. Notably, even though the predictor for Qwen3-8B is trained on reasoning traces generated by DeepSeek-R1, it still performs well. This suggests that the training data can be *model-agnostic*, provided the target LLM exhibits a similar reasoning style, for instance, using words like "wait" or "alternatively" to structure its reasoning process.

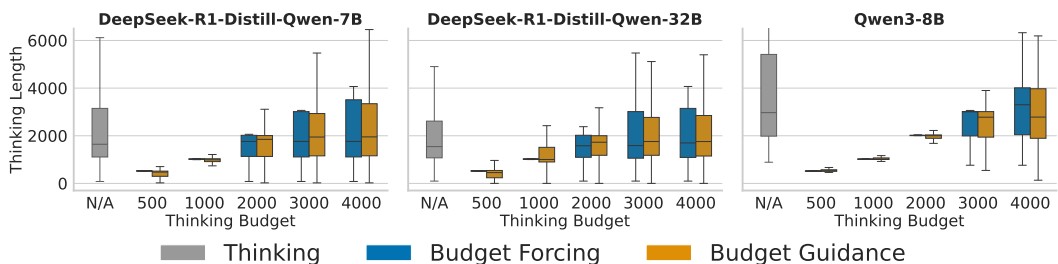

Figure 4: Thinking length controllability measured on MATH-500 benchmark.

### 4.2.2 ACCURACY–THINKING LENGTH TRADEOFF ANALYSIS

A key indicator of effective control is the ability to achieve higher accuracy under the same thinking length, which we call *token efficiency*. To evaluate and compare the token efficiency of our method across different reasoning lengths, we vary the token budget to obtain different average thinking lengths and record the corresponding accuracy achieved by the model. We visualize this relationship through accuracy-thinking length trade-off curves. Experiments are conducted on all three models across the four math benchmarks, and the resulting plots are presented in Figure 3.

From Figure 3, we observe that our method consistently achieves better token efficiency across most benchmarks, achieving higher accuracy than *budget forcing* under a range of thinking lengths. Notably, as the average thinking length decreases, corresponding to stricter budget constraints, our method yields significantly higher accuracy, particularly on benchmarks with diverse problem difficulty such as MATH-500. We attribute this to the ability of our method to adapt the reasoning pattern under strict budgets, producing concise yet complete reasoning traces. This enables the model to arrive at correct answers more efficiently, especially for questions that are relatively easy and do not require deep reasoning. This is also reflected in the occasional worse accuracy of *budget forcing* compared to the *NoThinking* baseline under strict budgets (*e.g.*, MATH-500 on DS-7B/32B), where the reasoning trace is abruptly truncated and the model is forced to guess prematurely. In contrast, our method avoids such incomplete reasoning and consistently outperforms the *NoThinking* baseline. Our method also performs better than Dynasor, another inference time steering method. We examine several failure cases of Dynasor and find that many stem from incomplete reasoning, even under its default settings. This further underscores the advantage of our method, which can precisely control reasoning length and thereby consistently deliver high-quality answers across different thinking budgets. An illustrative example of this guided reasoning behavior is provided in Appendix F.

### 4.2.3 FINE-GRAINED CONTROL OF THINKING LENGTH

Our goal is to steer LLM reasoning to adhere to a specified thinking budget. To evaluate controllability, we test on MATH-500 under varying thinking budgets, measuring the actual thinking length per sample and visualizing the distributions. We compare our method to *budget forcing* and include the full-thinking baseline as a reference. Results across all three models are shown in Figure 4.

From Figure 4, we observe that our method behaves similar to *budget forcing*, generally respects the specified thinking budget: for each setting, at least 75% are within the budget, and the median thinking length closely aligns with the budget. Compared to the full-thinking baseline, our method guides the model to generate a budget-aligned reasoning trajectory. This behavior is notable because, unlike *budget forcing*, our approach does not enforce a hard cutoff. Instead, it softly steers the generation process to match the desired level of detail, demonstrating flexible and controllable reasoning.

### 4.2.4 OUT-OF-DOMAIN TRANSFERABILITY

While we train the predictor solely on math data for simplicity, we also explore its generalization to broader task domains. To this end, we conduct an out-of-domain transferability analysis using the DS-7B model. Specifically, we evaluate our method on four benchmarks: **GPQA Diamond** (scientific reasoning), **FOLIO** (logical reasoning), **TableBench** (tabular reasoning), and **LiveCodeBench** (code

Table 2: Evaluation on out-of-domain transferability.

|  | GPQA Diamond | | FOLIO | | TableBench | | LiveCodeBench | |
| --- | --- | --- | --- | --- | --- | --- | --- | --- |
|  | Acc. | #Tokens | Acc. | #Tokens | Acc. | #Tokens | Acc. | #Tokens |
| Thinking | 49.1 | 5838 | 63.5 | 849 | 37.0 | 906 | 26.9 | 3509 |
| NoThinking | 38.4 | - | 46.3 | - | 16.9 | - | 20.7 | - |
| Budget Forcing | 39.9 | 1895 | 60.1 | 372 | 22.4 | **379** | 28.8 | **1135** |
| **Budget Guidance** | **49.0** | **1704** | **61.6** | **362** | **26.7** | 381 | **29.4** | 1138 |

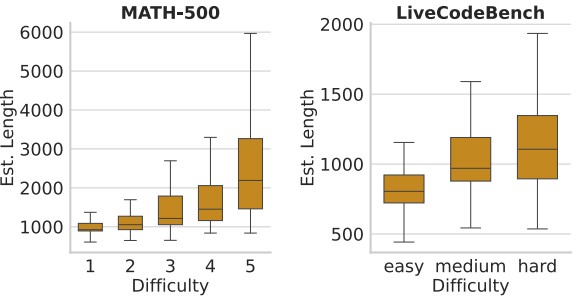

Figure 5: Correlation between question difficulties and estimated thinking lengths.

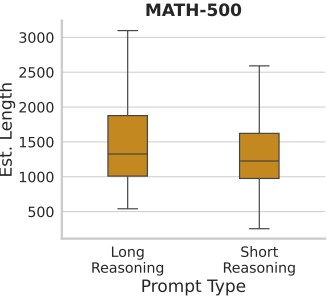

Figure 6: Correlation between prompt types and estimated thinking lengths.

reasoning). We match the average reasoning length between our method and the baseline, and report the corresponding accuracies in Table 2.

Despite being trained exclusively on math data, our predictor generalizes well to non-math reasoning tasks, consistently outperforming *budget forcing* across all four benchmarks. These results highlight the cross-domain generalizability of our approach and its potential applicability to a wide range of reasoning scenarios. While the gains on out-of-domain tasks are less pronounced than those on in-domain benchmarks, we believe performance can be further improved by incorporating reasoning traces from a broader range of domains during training. We leave this direction for future work.

### 4.3 Insights into What the Predictor Learns

To probe what the predictor has learned, we analyze its estimated thinking length at the first thinking token, interpreted as the predicted number of thinking tokens needed, against **task difficulty** and **prompt type**, using the DS-7B model.

**Task Difficulty.** We evaluate on MATH-500 (in-domain) and LiveCodeBench (out-of-domain). Figure 5 shows that estimated thinking length increases with difficulty in both cases. This suggests that the predictor captures a general understanding of difficulty, enabling effective difficulty estimation.

**Prompt Type.** We evaluate on MATH-500 and compare two prompts: one encouraging long reasoning and one encouraging concise reasoning (listed in the Appendix). As shown in Figure 6, the long reasoning prompt yields longer estimated thinking lengths. A t-test gives a $p$-value of 0.0028, confirming the difference is statistically significant and indicating that the predictor is prompt-aware.

Detailed behavioral analyses of the predictor, from both quantitative and qualitative perspectives, are presented in Appendix E and Appendix F, further illustrating how it steers reasoning length.

## 5 Conclusion

We introduce *budget guidance*, a simple yet effective approach for steering LLM reasoning under a thinking budget. Without requiring any LLM fine-tuning, our method enables natural control over the reasoning process and significantly improves token efficiency on challenging benchmarks. Our results demonstrate that LLMs can be effectively guided to reason with *budget guidance*, highlighting budget-conditioned generation as a promising direction for efficient and controllable LLM reasoning.

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

# A  PREDICTOR LATENCY OVERHEAD ANALYSIS

To better quantify the latency overhead introduced by our predictor, we conduct precise measurements on three representative models using the HuggingFace `transformers` library on a single NVIDIA H100 GPU. We generate 1000 thinking tokens per run on the full AIME-2024 test set, with each question executed 10 times to compute an average latency. Final results are averaged across questions, and we report 95% confidence intervals for clarity.

Table 3: Measured latency overhead of our predictor across different LLMs.

| Model | Latency Overhead | 95% Confidence Interval |
|---|---|---|
| DeepSeek-R1-Distill-Qwen-7B | 0.72% | [0.60%, 0.83%] |
| DeepSeek-R1-Distill-Qwen-32B | 0.17% | [0.12%, 0.21%] |
| Qwen3-8B | 0.48% | [0.27%, 0.68%] |

As shown in Table 3, the latency overhead for all three models is below 0.8%, and decreases further for larger models (only 0.17% for a 32B LLM). This demonstrates that our predictor introduces negligible additional cost relative to standard inference, and thus does not compromise the efficiency of the overall system.

# B  ABLATION ON SKIPPING STRATEGIES

We further conduct an ablation study to examine different strategies for applying budget guidance. We consider three settings:

- Apply the modulation at the beginning of every sentence.
- Apply the modulation at every token (no skipping).
- Apply the modulation at the beginning of every paragraph (this is the strategy used in our main paper).

We evaluate these strategies on the MATH-500 benchmark using the DeepSeek-R1-Distill-Qwen-7B model. Results are summarized in Table 4.

Table 4: Ablation study on different skipping strategies for applying budget guidance.

| Strategy | Accuracy (%) | #Tokens |
|---|---|---|
| Apply at beginning of sentence | 88.0 | 1333 |
| Apply at every token | 86.0 | 1448 |
| Apply at beginning of paragraph | **88.2** | **1329** |

We observe that there is no significant difference between applying guidance at the beginning of a sentence or a paragraph. Since the paragraph-level strategy achieves comparable accuracy while saving more computation, we adopt it in our main experiments. Interestingly, applying guidance at every token leads to lower accuracy. Closer inspection shows that guidance applied mid-sentence is less stable, often disrupting semantic coherence. This instability can accumulate across token generations and result in random or repeated outputs. For both inference stability and efficiency, we therefore do not recommend modulating every token.

# C  ABLATION ON PREDICTOR ARCHITECTURE

We also examine whether a more lightweight alternative to the BERT-based length predictor is feasible. Specifically, we remove the BERT encoder and use only a single linear layer to predict the parameters of the Gamma distribution. We apply the same token budget to this variant and evaluate its performance on the MATH-500 benchmark.

Table 5: Comparison of predictor architectures on the MATH-500 benchmark.

| Predictor Model | MATH-500 Acc. | #Tokens |
|---|---|---|
| Linear Layer | 85.8 | 1617 |
| BERT | **88.2** | **1329** |

As shown in Table 5, the BERT-based predictor clearly outperforms the simple linear layer. In particular, the linear layer is less effective in controlling reasoning length, as evidenced by its longer outputs compared to the BERT-based predictor under the same budget. This suggests that estimating the remaining reasoning length is a challenging task, and a larger model such as BERT offers stronger predictive capacity. There exists a trade-off between predictor accuracy and efficiency; in this paper, we adopt the BERT-based predictor due to its robust performance and widespread use across many tasks.

## D    ANALYSIS OF INTERVENTIONS ON SAMPLES WITH DIFFERENT REASONING LENGTH REQUIREMENTS

To further understand the behavior of Budget Guidance, we analyze its effect on samples requiring shorter versus longer reasoning traces. We compute the relative percentage change in reasoning length on the MATH-500 dataset using the DeepSeek-R1-Distill-Qwen-7B model, defined as

$$\Delta = \frac{|\text{length}_{\text{vanilla}} - \text{length}_{\text{BG}}|}{\text{length}_{\text{vanilla}}}.$$

Table 6: Length change for samples with different reasoning requirements on MATH-500.

| Sample Category | Change in Length |
|---|---|
| Short reasoning | 15.3% |
| Long reasoning | 52.3% |

We find that Budget Guidance more significantly reduces reasoning length for samples originally above the budget, while having minimal effect on those below. Notably, 100% of samples already under the budget remain correct, indicating no accuracy loss in these cases.

## E    QUANTITATIVE REASONING BEHAVIOR ANALYSIS

To quantitatively analyze how the predictor influences the reasoning behavior of LLMs under different budget settings, we follow the methodology proposed by (Hou et al., 2025). Specifically, we count the frequency of reasoning-related keywords such as *"wait"* and *"alternatively"*, which are indicative of deeper reasoning processes. We compare the keyword frequencies for thinking budget of 500, 2000, and 4000 tokens using the DS-7B model on the MATH-500 benchmark. These results are contrasted with a full-thinking baseline (*i.e.*, without applying our method). The comparison is illustrated in Figure 7.

As shown in the figure, a smaller budget substantially reduces the frequency of reasoning-related keywords, indicating a more concise reasoning process. As the budget increases, the model is encouraged to engage in deeper reasoning. Notably, when the budget is set sufficiently high, the behavior closely matches that of the full-thinking baseline, suggesting minimal loss in reasoning capability. These findings demonstrate that our method can effectively steer the reasoning behavior of LLMs, while still preserving their reasoning ability under higher budget constraints.

## F    QUALITATIVE REASONING BEHAVIOR ANALYSIS

Figure 8 shows a case study from MATH-500 illustrating reasoning traces under different thinking budgets. Rather than truncating output, our method adapts the reasoning style to the budget. With a

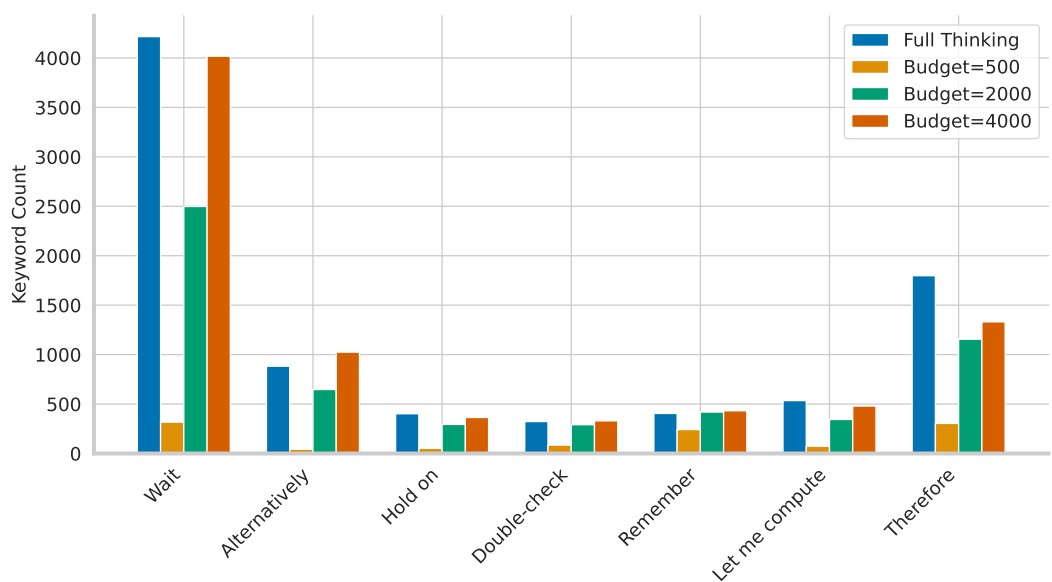

Figure 7: Reasoning keywords frequency comparison under different budget settings.

stricter budget (left), the model generates concise answers without reflection. With a larger budget (right), it mirrors full-length reasoning: it starts with problem analysis and using reflective phrases like "wait" and "double-checking." In both settings, the trace ends appropriately, highlighting our method's flexibility and controllability.

---

**Question:** The graph of $f(x) = \frac{2x}{x^2 - 5x - 14}$ has vertical asymptotes $x = a$ and $x = b$, and horizontal asymptote $y = c$. Find $a + b + c$.

---

**Thinking 1 (budget = 300)**

To determine the vertical asymptotes of the function $f(x) = \frac{2x}{x^2 - 5x - 14}$, I first need to find the values of $x$ that make the denominator zero, as these will be the points where the function is undefined and thus where the vertical asymptotes occur.

I'll start by factoring the denominator:

$$x^2 - 5x - 14 = (x - 7)(x + 2)$$

Setting each factor equal to zero gives the solutions $x = 7$ and $x = -2$. Therefore, the vertical asymptotes are at $x = 7$ and $x = -2$. Next, to find the horizontal asymptote, I'll compare the degrees of the numerator and the denominator. The numerator has a degree of 1, and the denominator has a degree of 2. Since the degree of the denominator is higher, the horizontal asymptote is $y = 0$. Finally, I'll add the values of $a$, $b$, and $c$:

$$a + b + c = 7 + (-2) + 0 = 5$$

**(thinking length: 260)**

**Thinking 2 (budget = 600)**

**Okay,** so I have this function here: f(x) = (2x)/(x² - 5x - 14). I need to find the vertical asymptotes and the horizontal asymptote, and then add their x-values and the y-value together to get a + b + c. **Hmm,** let me think about how to approach this.

First, vertical asymptotes occur where the denominator is zero, right? Because the function will be undefined at those points, and the graph will shoot off to infinity or negative infinity. So, I need to factor the denominator to find those x-values.

The denominator is x² - 5x - 14. Let me try to factor that. I'm looking for two numbers that multiply to -14 and add up to -5. **Hmm, let's see.** The factors of 14 are 1 and 14, 2 and 7. Since the product is negative, one has to be positive and the other negative. Let me try 2 and -7: 2 * (-7) = -14, and 2 + (-7) = -5. Perfect! So, the denominator factors as (x + 2)(x - 7). So, the denominator is zero when x + 2 = 0 or x - 7 = 0, which means x = -2 and x = 7. Therefore, the vertical asymptotes are at x = -2 and x = 7. So, a and b are -2 and 7, respectively.

**Wait, hold on,** the problem says vertical asymptotes x = a and x = b. So, a is -2 and b is 7? Or is it the other way around? **Hmm,** actually, the order doesn't matter because we're just adding them together. So, a + b will be (-2) + 7, which is 5. Okay, that's straightforward.

Now, onto the horizontal asymptote. Horizontal asymptotes depend on the degrees of the numerator and the denominator. The numerator is 2x, which is degree 1, and the denominator is x² - 5x -14, which is degree 2. So, the degree of the numerator is less than the degree of the denominator. I remember that when the degree of the numerator is less than the degree of the denominator, the horizontal asymptote is y = 0. So, c is 0.

**Double-checking:** If the degree of the numerator is less than the denominator, horizontal asymptote is y = 0. If they are equal, it's the ratio of the leading coefficients. If the numerator is higher, there's no horizontal asymptote, but maybe an oblique one. So, in this case, since numerator is degree 1 and denominator is degree 2, horizontal asymptote is y = 0. So, c = 0.

**(thinking length: 602)**

Figure 8: Sample reasoning traces generated with *budget guidance* under different thinking budgets.

## G   DATASET DESCRIPTION

We provide detailed information about the evaluation datasets used in our paper.

**MATH-500** (Hendrycks et al., 2021) is a 500-problem subset of the MATH dataset, selected by (Lightman et al., 2023). Each problem is labeled with a difficulty level from 1 to 5.

**AIME-2024** (Art of Problem Solving, n.d.a) contains 30 problems from the 2024 American Invitational Mathematics Examination, covering topics such as algebra, combinatorics, geometry, number theory, and probability. Following *budget forcing* (Muennighoff et al., 2025), we retain only the essential ASY figure code required to solve each problem, omitting non-essential diagrams.

**AMC** (Art of Problem Solving, n.d.b) includes all 83 problems from AMC12 2022 and AMC12 2023.

**OlympiadBench** (He et al., 2024) is a challenging benchmark aimed at advancing AGI through Olympiad-level, bilingual, multimodal scientific problems. We use its math subset, which contains a total of 675 problems.

**GPQA Diamond** (Rein et al., 2024) consists of 198 high-quality, extremely difficult questions spanning a broad range of scientific domains, including biology, physics, and chemistry.

**FOLIO** (Han et al., 2022) is a human-annotated dataset designed to evaluate complex logical reasoning in natural language. It features 1,430 unique conclusions paired with 487 sets of premises, all validated using first-order logic (FOL) annotations. We use the test set, which contains 203 unique problems.

**TableBench** (Wu et al., 2025) is a benchmark for evaluating LLMs on real-world tabular data challenges. We evaluate all models on the numerical reasoning subset, which comprises 493 problems.

**LiveCodeBench** (Jain et al., 2024) offers a holistic and contamination-free evaluation of LLM coding capabilities. Following (Guo et al., 2025), we select problems from the August 2024 to January 2025 period, totaling 323 problems.

## H    TRAINING DATA AUGMENTATION

We adopt a simple data augmentation strategy to double the size of the training set. Each training sample originally follows the format:

$$\texttt{<think>THINK\_MESSAGE</think>ANSWER\_MESSAGE} \tag{8}$$

Since our predictor only operates on the `THINK_MESSAGE`, the `ANSWER_MESSAGE` is not used during training. To utilize this otherwise unused information, we construct an additional training sample in the following format:

$$\texttt{<think>ANSWER\_MESSAGE</think>ANSWER\_MESSAGE} \tag{9}$$

This transformation allows us to incorporate the `ANSWER_MESSAGE` into the predictor's training process. By generating one new sample for each original sample, we effectively double the size of the training set and ensure full utilization of the available data.

## I    PROMPT DESCRIPTION

In Section 4.3, we analyze the predictor's estimated thinking length across different prompt types to demonstrate its prompt awareness. Below, we list the specific prompts used in our experiment.

The prompt for long reasoning is: `Think step by step and provide thorough reasoning before reaching a conclusion.`

The prompt for short reasoning is: `Think quickly and provide a concise reasoning with minimal steps.`

We add these prompts as the system prompt.

## J    USE OF LARGE LANGUAGE MODELS FOR WRITING ASSISTANCE

Portions of the writing in this paper, specifically at the level of grammar refinement, sentence polishing, and shortening of paragraphs for conciseness, were assisted by an external large language

model (OpenAI ChatGPT). The model was not used to generate original ideas, experimental design, or analysis; all scientific contributions are the authors' own. The assistance was limited to improving clarity, readability, and presentation quality of the manuscript.

