# OpenReview forum: "Steering LLM Thinking with Budget Guidance"
_ICLR.cc/2026/Conference — ICLR 2026 Conference Withdrawn Submission_

### Official Review · Reviewer_x6w8 · 2025-10-22

**Soundness:** 2
**Presentation:** 3
**Contribution:** 2
**Rating:** 2
**Confidence:** 4

**Summary:**

This paper proposes a budget guidance method for steering the reasoning process of LLMs without requiring any LLM fine-tuning. Our approach introduces a lightweight predictor that models a Gamma distribution over the remaining thinking length during next-token generation. This signal is then used to guide generation in a soft, token-level manner, ensuring that the overall reasoning trace
adheres to the specified thinking budget. Budget guidance achieves up to a 26% accuracy gain on the MATH-500 benchmark under tight budgets compared to baseline methods, while maintaining competitive accuracy with only 63% of the thinking tokens used by the full-thinking model.

**Strengths:**

This paper proposes a budget guidance method for steering the reasoning process of LLMs without requiring any LLM fine-tuning. It can achieve up to a 26% accuracy gain on the MATH-500 benchmark under tight budgets compared to baseline methods, while maintaining competitive accuracy with only 63% of the thinking tokens used by the full-thinking model. Different benchmarks have been tested.

**Weaknesses:**

1. The method's core relies on modeling the remaining thinking length with a Gamma distribution. This is a strong assumption about the nature of the reasoning process.
2. The paper compares against "baseline methods," but it's important to define these and include more sophisticated alternatives. So many papers have discussed on overthinking and underthinking. More experiments should be added.

**Questions:**

More experiments should be added.

---

> ### Comment · Reviewer_x6w8 · 2025-11-26
> **Update the score**
>
> I understand the added experiments and I update my score to 4.

---

> ### Author Response · Authors · 2025-11-26
>
> > 1. The method's core relies on modeling the remaining thinking length with a Gamma distribution. This is a strong assumption about the nature of the reasoning process.
>
>
> Thank you for raising this point. We would like to clarify that **using a Gamma distribution does not impose a strong assumption on the reasoning dynamics**. The distribution is employed solely to model the *remaining length*, a strictly positive variable similar to waiting time in well-known duration and time-to-event modeling, for which the Gamma distribution is a classical, lightweight, well-behaved, and widely used positive-support family. Our method does not assume that the full reasoning trajectory follows a Gamma process; the predictor simply fits a two-parameter approximation to this scalar (the remaining reasoning length) quantity at each step as an efficient surrogate for much more complex behavior. This choice yields stable estimates with negligible overhead, whereas more expressive alternatives, like nonparametric or mixture models, would potentially increase inference cost or add many more parameters, making training and inference harder. Thus, the Gamma distribution serves as a practical approximation for remaining-length prediction, not as a strong assumption about how reasoning itself is generated.
>
> > 2. The paper compares against "baseline methods," but it's important to define these and include more sophisticated alternatives. So many papers have discussed on overthinking and underthinking. More experiments should be added.
>
> The area of efficient reasoning is moving fast, so we only compare to those baselines that, at the time we are writing the submission, are proved to be effective and widely used in industry and billion user applications. Nevertheless, we are fully willing to incorporate additional baselines to further demonstrate the effectiveness of our approach, and we provide more baseline results on G1 in General Response. In summary, our method is generally better than all these baselines, providing a strong, unique and elegant solution for LLM budget conditioned generation.

---

### Official Review · Reviewer_Xrsi · 2025-10-27

**Soundness:** 3
**Presentation:** 4
**Contribution:** 2
**Rating:** 4
**Confidence:** 5

**Summary:**

The paper proposes Budget Guidance, a test-time approach for steering the reasoning process of large language models without requiring any fine-tuning. The method employs a lightweight predictor to estimate the remaining reasoning length and uses this prediction to dynamically guide the generation process.

The central insight is that the budget-conditional distribution can be derived from the unconditional distribution combined with the predicted remaining length distribution, allowing the model to adjust its reasoning behavior at the beginning of each reasoning segment.

Experimental results demonstrate that Budget Guidance consistently outperforms Budget Forcing while producing responses with comparable or shorter reasoning lengths.

**Strengths:**

1. The paper introduces a fine-tuning–free approach to guide the reasoning process of large language models (LLMs), which can be seamlessly integrated as a plugin into modern LLM serving systems.
2. The proposed Budget Guidance mechanism applies a soft, token-level steering that effectively regulates reasoning length while preserving the model’s inherent reasoning capability.
3. The predictor operates only once at the start of each reasoning segment, resulting in negligible latency overhead and maintaining system efficiency during inference.

**Weaknesses:**

1. Limited baselines: The paper only compares against one budget-based method. Missing broader hybrid-thinking or trade-off baselines, such as prompt-based classification or adaptive fast/slow reasoning strategies.
2. Lack of diversity in model structure: All tested models (DeepSeek-R1-7B/32B, Qwen3-8B) are dense models. It’s unclear whether Budget Guidance remains effective on MoE architectures.
3. Single token budget objective setting: Evaluations only target at half the original model’s full thinking length, without exploring adaptive or partial-budget variants.

**Questions:**

1. The paper does not report the number of NoThinking tokens, which prevents a clear and quantitative comparison between the proposed approach and baseline methods.
2. The predictor for Qwen3-8B is trained on reasoning traces from DeepSeek-R1 yet performs well. The authors claim model-agnostic generalization—could this result instead arise from architectural and scale similarity between dense models of 8B–32B size? (As the base model of DeepSeek-R1-Distill-Qwen is still Qwen)
3. The thinking budget is set to approximately half of the full reasoning length. What is the rationale behind this choice? Have other budget ratios been tested or compared?
4. Appendix H mentions that the predictor’s training data includes sequences like `<think>ANSWER_MESSAGE</think>ANSWER_MESSAGE`. What is the motivation for including such patterns, and could this design introduce misleading supervision signals?
5. Typo: Page 4, Line 287 — “OpenR1-Math-220k (Face, 2025)” should be corrected to “OpenR1-Math-220k (Hugging Face, 2025)”.

---

> ### Author Response · Authors · 2025-11-26
>
> > 1. Limited baselines: The paper only compares against one budget-based method. Missing broader hybrid-thinking or trade-off baselines, such as prompt-based classification or adaptive fast/slow reasoning strategies.
>
> Please see G1 in General Response for the comparison with more baselines.
>
> > 2. Lack of diversity in model structure: All tested models (DeepSeek-R1-7B/32B, Qwen3-8B) are dense models. It’s unclear whether Budget Guidance remains effective on MoE architectures.
>
>
> We are happy to provide additional experiments on MoE architectures, so we train the predictor for Qwen3-30B-A3B model, which is a MoE model with 30B total parameters and 3B activated parameters. We evaluate on MATH-500 benchmark with three thinking budget levels. The results are in the table below.
>
> |Qwen3-30B-A3B| Accuracy | # Thinking Tokens |
> |---|---|---|
> | Full Thinking Baseline | 96.8 | 4315 |
> | *Low Thinking Budget* |  |  |
> | Budget Forcing | 86.0 | 1058 |
> | **Budget Guidance** | **91.8** | 969 |
> | *Medium Thinking Budget* |  |  |
> | Budget Forcing | 88.8 | 1650 |
> | **Budget Guidance** | **92.6** | 1605 |
> | *High Thinking Budget* |  |  |
> | Budget Forcing | 93.8 | 3405 |
> | **Budget Guidance** | **94.6** | 3202 |
>
> Similar to dense model, our method continues to work with MoE model, and performs better than budget forcing. This is not surprising as our method does not impose any restriction on the model architectures and should work properly on most, if not any, LLMs.
>
> > 3. Single token budget objective setting: Evaluations only target at half the original model’s full thinking length, without exploring adaptive or partial-budget variants.
> > The thinking budget is set to approximately half of the full reasoning length. What is the rationale behind this choice? Have other budget ratios been tested or compared?
>
> We do evaluate our method across a wide range of budgets, as shown in Figure 3 of the main paper, where accuracy–token tradeoffs are reported for multiple budget ratios. The choice of approximately half of the full reasoning length in Table 1 is mainly for comparability, as it allows all baselines to operate under a similar token budget, making their accuracy directly comparable. In contrast, our full curves in Figure 3 already include both smaller and larger budget ratios, demonstrating that our method behaves consistently across different budget levels.
>
> > 4. The paper does not report the number of NoThinking tokens, which prevents a clear and quantitative comparison between the proposed approach and baseline methods.
>
> NoThinking method, as its name suggests, does not have any thinking process and has only one fixed sentence ("Okay, I think I have finished thinking") in the thinking part to force the model to bypass thinking, so we can treat its thinking token count as **zero**. NoThinking is generally served as a potential lower bound for thinking methods, as it does not think at all. We will clarify this in our revision.
>
>
> > 5. The predictor for Qwen3-8B is trained on reasoning traces from DeepSeek-R1 yet performs well. The authors claim model-agnostic generalization—could this result instead arise from architectural and scale similarity between dense models of 8B–32B size? (As the base model of DeepSeek-R1-Distill-Qwen is still Qwen)
>
> We do not intend to make a strong claim of model-agnostic generalization. Instead, we view the observed transferability as a consequence of the data-driven nature of our method. Many reasoning-focused LLMs exhibit broadly similar stylistic patterns, for example, using phrases such as “wait” or “alternatively” when elaborating on longer chains of thought, and employing shorter equations or more concise formulations when reasoning briefly. Our approach does not impose any of these patterns, nor does it force the model to reason longer or shorter. Rather, these tendencies are implicitly captured by the predictor through the training data, and the predictor then steers the reasoning in a purely data-driven manner.
>
> This explains why a predictor trained on DeepSeek-R1 reasoning traces can still generalize to Qwen3-8B model: both models share similar observable reasoning behaviors. Conversely, if a model were to reason in a fundamentally different manner, we would not expect a predictor trained on DeepSeek-R1 traces to transfer effectively. Thus, we do not claim universal model-agnostic generalization; the degree of transferability depends on the alignment of reasoning patterns across models.

---

> ### Author Response · Authors · 2025-11-26
> **Official Comment by Authors (cont'd)**
>
> > 6. Appendix H mentions that the predictor’s training data includes sequences like <think>ANSWER_MESSAGE</think>ANSWER_MESSAGE. What is the motivation for including such patterns, and could this design introduce misleading supervision signals?
>
> As explained in the Appendix, the main motivation was to fully leverage the available training data. In fact, our subsequent experiments show that including this component does not significantly affect performance, so we moved it to the supplementary and marked it as an optional data augmentation strategy in the revision.
>
> > 7. Typo: Page 4, Line 287 — “OpenR1-Math-220k (Face, 2025)” should be corrected to “OpenR1-Math-220k (Hugging Face, 2025)”.
>
> Thanks for pointing this out. We will fix this in our revision.

---

### Official Review · Reviewer_FiiM · 2025-10-31

**Soundness:** 3
**Presentation:** 2
**Contribution:** 2
**Rating:** 4
**Confidence:** 4

**Summary:**

This paper presents an adaptive LLM thinking method that uses a budget-conditional output distribution to control the output length under a budget-constrained setting. Through a Bayesian perspective, it models the conditional likelihood of the remaining length distribution using a Gemma distribution, which can easily produce the cumulative probability of length. By collecting reasoning chain data, it trains a predictor using MLE to fit two intrinsic parameters in the Gemma distribution, where the parameterization is based on a BEART model and embeddings of the target LLM. Empirical results show that the proposed method achieves comparable results with the thinking baseline with less tokens, and achieves higher performance than the Budget Forcing baseline.

**Strengths:**

- The proposed budget-conditional output distribution is interesting, and the framework has a principled design.
- The trained predictor shows task generalization, even though it is only trained on math.
- The paper is easy to understand.

**Weaknesses:**

- The evaluated baselines only include NoThinking and Budget Forcing, which are not representative and strong enough. There are many other existing adaptive thinking methods, but the authors did not review or compare their method against other adaptive thinking methods. This makes the empirical evaluation of the proposed method weak. I would suggest the other at least compare with two other adaptive thinking methods to show the real effective of the proposed method.
- If taking the last token embedding of all layers, it would be very costly to train such a predictor. Some inference cost analysis is shown in Table 3 for the proposed method, but the training overhead is unclear. It would be clearer if the author could show the training cost and compare the variant that only trains on the last token embedding on the last layer.
- The choice of the Gemma distribution is not justified. Is there any other distribution that is also suitable for modeling the length distribution? Can the author give justification/explanation and comparison with other distributions?
- The task generalization only shows that the math data could help to generalize to other domains. It is unclear whether the predictor trained on another domain could generalize to math or not. If it is not, its application would be limited if no high-quality math reasoning data is available.

**Questions:**

- In line 265, is this “linear projection” trainable or a fixed projection at initialization? Could you explain more about this? Is the [CLS] token also used as part of the input?

---

> ### Author Response · Authors · 2025-11-26
>
> > 1. The evaluated baselines only include NoThinking and Budget Forcing, which are not representative and strong enough. There are many other existing adaptive thinking methods, but the authors did not review or compare their method against other adaptive thinking methods. This makes the empirical evaluation of the proposed method weak. I would suggest the other at least compare with two other adaptive thinking methods to show the real effective of the proposed method.
>
> The area of efficient reasoning is very active and moving fast, and we are happy to add several SOTA baselines for a more comprehensive comparison. We provide three more baselines, in total six baselines across three models and four benchmarks, to show the real effective of the proposed method. Please see the G1 in General Response for the full evaluation results.
>
>
> > 2. If taking the last token embedding of all layers, it would be very costly to train such a predictor. Some inference cost analysis is shown in Table 3 for the proposed method, but the training overhead is unclear. It would be clearer if the author could show the training cost and compare the variant that only trains on the last token embedding on the last layer.
>
>
> The variant that only uses the last-layer, last-token embedding is already included in Table 5 of Appendix C. As shown there, relying solely on the final layer leads to a clear drop in performance, suggesting that the multi-layer representation contains additional signals that are useful for predicting the remaining reasoning length.
>
> Regarding the training cost, the overhead is lightweight because the base LLM remains frozen and the predictor is a small BERT encoder with a linear head. The training process consists of two stages: first, we run batched forward passes on the frozen LLM to extract layerwise hidden states; second, we train the BERT predictor on these extracted features. We profiled the full pipeline using DeepSeek-R1-Distill-Qwen-7B model with FlashAttention on a single H100 GPU. In general, extracting hidden states takes about 25 ms for each sample, while training the BERT encoder takes 65 ms per sample. With a total of 220k samples in our training data, the end-to-end training time is approximately 5.5 hours. Given that the LLM is never updated and all computation is simple batched inference or small-model training, the overall cost is modest compared to any fine-tuning–based approach.
>
>
> > 3. The choice of the Gemma distribution is not justified. Is there any other distribution that is also suitable for modeling the length distribution? Can the author give justification/explanation and comparison with other distributions?
>
> Please see G2 in General Response for the reason why using Gamma distribution.
>
>
> > 4. The task generalization only shows that the math data could help to generalize to other domains. It is unclear whether the predictor trained on another domain could generalize to math or not. If it is not, its application would be limited if no high-quality math reasoning data is available.
>
> Thank you for your comment. We have added an experiment where the predictor is trained on code data (the code split from Mixture-of-Thoughts dataset) and evaluated on MATH-500 benchmark. the results are shown in the table below.
>
> | DeepSeek-R1-Distill-Qwen-7B | MATH-500_Accuracy | MATH-500_#Tokens |
> |---|---|---|
> | train on math data | 88.2 | 1329 |
> | train on code data | 88.6 | 1409 |
>
>
> We can see that the predictor trained on code data can also generalize to math. While we do not intend to claim strong task generalization, we believe that as long as the model exhibits similar reasoning patterns across different tasks, it can still be effective to some extent.
>
> > 5. In line 265, is this “linear projection” trainable or a fixed projection at initialization? Could you explain more about this? Is the [CLS] token also used as part of the input?
>
> The linear projection is effectively a trainable linear layer. The [CLS] token is part of the original BERT architecture, where it is used to aggregate information from all input tokens. We do not modify this component in our approach.

---

### Official Review · Reviewer_qW8j · 2025-10-31

**Soundness:** 4
**Presentation:** 4
**Contribution:** 3
**Rating:** 8
**Confidence:** 4

**Summary:**

This paper presents a novel inference-time method called "budget guidance" for controlling the reasoning length of large language models (LLMs) without requiring fine-tuning. The approach introduces a lightweight auxiliary predictor that estimates the remaining thinking length using a Gamma distribution, which is then used to softly steer token-level generation toward a specified budget. Experiments on math reasoning benchmarks demonstrate significant improvements in token efficiency, with maintained accuracy under tight budgets, and the method shows promising cross-domain generalization. The work also provides insights into the predictor's ability to infer task difficulty and respond to prompt-based instructions.

**Strengths:**

1）The proposed “budget guidance” is a simple yet effective inference-time method that enables flexible control over chain-of-thought (CoT) length without supervised fine-tuning (SFT), avoiding computational costs and potential safety risks associated with LLM fine-tuning.

2）The lightweight predictor is innovative, as it models remaining reasoning length distributionally and generalizes well to out-of-domain tasks (e.g., scientific and logical reasoning), highlighting its strong adaptability.

3）The method enhances token efficiency substantially, achieving competitive accuracy with significantly shorter reasoning traces.

Overall speaking, the key originality lies in implicitly guiding LLMs based on remaining length predictions, avoiding hard cut-offs or rule-based interventions while preserving reasoning quality.

**Weaknesses:**

The study lacks a prompt engineering (PE) baseline. Given the insights that the predictor is prompt-aware and capable of adapting to instructions (e.g., generating concise or lengthy reasoning), a comparison with well-crafted prompts could validate whether similar length-control effects can be achieved without the predictor, strengthening the claim of necessity.

**Questions:**

Why is Dynasor not included as a baseline in Table 1? As an inference-time steering method, Dynasor's performance under comparable settings (e.g., token budgets) would offer a more comprehensive evaluation relative to existing baselines like budget forcing and NoThinking.

---

> ### Author Response · Authors · 2025-11-26
>
> > prompt engineering (PE) baseline
>
> Thanks for the suggestion. We are happy to provide two prompt engineering baselines, namely Chain-of-Draft [1] and TALE-EP [2], which utilize specialized and well-crafted prompts to shorten the reasoning length. Please refer to the G1 in General Response for the full results. In summary, our method is generally better than prompt-based methods, without the need to worry about the hassle of prompt engineering. Also, through our experiment, we observed that prompt-based methods do not control the reasoning length effectively, and tricks like including the set budget in the prompt (which is what TALE-EP does) do not guide the model to follow these budgets effectively for the three models that we evaluated.
>
>
> > Why is Dynasor not included as a baseline in Table 1.
>
> Thank you for pointing this out. Dynasor is indeed a relevant inference-time steering method, and we include Dynasor in Table 1 in our revision.

---

### Official Review · Reviewer_av2G · 2025-11-01

**Soundness:** 3
**Presentation:** 2
**Contribution:** 2
**Rating:** 2
**Confidence:** 4

**Summary:**

This paper proposes a budget-aware controllable LLM next-token-prediction for efficient thinking. In which, the final token distribution is modified with the learned predictor score distribution using a lightweight BERT model. The proposed Budget guidance does not need to train the main LLM but only to train the auxiliary module to enable test-time control over the reasoning length of LLMs.

The proposed method is novel, but the additional n-dimensional vector $a_t$ is not clearly defined. In addition, how and why the Gemma modeling of remaining length is used to affect the final token distribution with $c_t$ are not well explained, making the method not so convincing.

Many training-free efficient test-time scaling methods are proposed, which are not compared as baselines. The accuracy loss of the proposed method on AIME24, AMC, and TableBench is still noticeable. Therefore, it is hard to validate the effectiveness of this paper.

**Strengths:**

This paper proposes a budget-aware controllable LLM next-token-prediction for efficient thinking. The authors propose to directly model the remaining thinking length as a Gamma distribution and use the CDF of Gamma distribution to compute the predictor score, which are used to steer the final LLM token distribution. The normalized product of $u_t$ and $a_t$ is defined as the conditional vector.

**Weaknesses:**

The underlying motivations of using the Gamma distribution to model the remaining length and using the CDF of it as the generated steering vectors to guide the final output decoding of LLMs are not clearly explained.

The experiments are not thoroughly conducted with more missing SOTA training-free baselines. In addition, the accuracy degradation of three datasets are not ignorable.

The end-2-end runtime reduction and the additional cost of embedding and BERT inference are not reported.

**Questions:**

See the weaknesses.

---

> ### Author Response · Authors · 2025-11-26
>
> > 1. The underlying motivations of using the Gamma distribution to model the remaining length and using the CDF of it as the generated steering vectors to guide the final output decoding of LLMs are not clearly explained.
>
>
>
> Thank you for raising this question. We are happy to clarify both the choice of the Gamma distribution and the use of its CDF during decoding. Please refer to the G2 in General Response for the reason why we use Gamma distribution.
>
> As for why we use the CDF, in our formulation, the decoding rule in Eq. (1) requires the probability
> $$
> P(L_t \le \bar{l} - t \mid \text{context}, Y_t=v),
> $$
> i.e., the probability that the remaining reasoning length is *within* the available budget. By definition, this probability is the integral of the remaining-length density from 0 to $\bar{l} - t$. The cumulative distribution function of the predicted Gamma distribution computes exactly this quantity:
> $$
> \mathrm{CDF}(\bar{l} - t) = \int_0^{\bar{l}-t} p(L_t = \ell)\, d\ell.
> $$
>
> Thus, using the CDF is not a heuristic choice, instead it is the mathematically correct way to obtain the probability term required for budget-conditional decoding.
>
> We would provide a clear clarification in our revision.
>
>
>
> > 2. The experiments are not thoroughly conducted with more missing SOTA training-free baselines. In addition, the accuracy degradation of three datasets are not ignorable.
>
>
> Thank you for the comment. The area of efficient reasoning has evolved rapidly, and we are happy to add several SOTA baselines for a more comprehensive comparison. Please see the G1 in General Response for details.
>
> For the accuracy degradation, as the saying goes, **there is no free lunch**. We acknowledge that enforcing a thinking-token budget naturally leads to some accuracy reduction, as the model is required to produce shorter reasoning traces than in its full-thinking setting, especially under a tight budget. The observed degradation for our method is moderate, expected, and much smaller than that of all other training-free baselines. We therefore believe the accuracy behavior is both principled and fully aligned with the goal of budget-controlled reasoning.
>
>
> > 3. The end-2-end runtime reduction and the additional cost of embedding and BERT inference are not reported.
>
> Thank you for the comment. The additional cost of the embedding extraction and the BERT-based predictor is already analyzed in Table 3 of Appendix A, which shows that their overhead remains below 1% for all evaluated models. Following other efficient reasoning baselines and literature, we do not directly report the end-to-end runtime reduction, as it varies greatly depending on the implementation and workload patterns. However, one can reasonably expect the runtime reduction to be roughly the same as the generated token reduction, as an auto-regressive LLM generates tokens at a constant speed.

---

> > ### Comment · Reviewer_av2G · 2025-11-28
> > **Response to Authors**
> >
> > It makes sense to model the remaining time or generation length using time-to-event analysis and then use the Gemma distribution. However, does the remaining generation length actually match the assumptions of Gemma distribution? Could you also provide some empritical statistical study of it?
> >
> > Although there is no free lunch, reasoning is originally designed to improve the accuracy and the 10~20% accuracy loss is hard to verify the effectiveness of the proposed method.  How the thinking budget is set in Table 1? Is it a input argument of the BERT model?
> >
> > I do agree that the final multi-batch reasoning inference acceleration requires some engineering works and better system design. Many released papers in arxiv only report token reduction without practical runtime reduction, but it is critical for the  efficiency improvement target.

---

### Author Response · Authors · 2025-11-26
**General Response**

We sincerely thank all reviewers for their valuable comments and suggestions. Below, we provide a general response addressing the most common concerns.

> G1. Comparison with more baselines。

The area of efficient reasoning has evolved rapidly, and we are happy to add several SOTA baselines for a more comprehensive comparison. Apart from the three baselines presented in our original submission, we additionally include the following three baselines:

- chain-of-draft [1]: a training-free prompting method that asks the LLM to generate minimal, highly compressed intermediate “draft” thoughts instead of verbose CoT.
- TALE-EP [2]: a training-free prompting method that includes the budget in the prompt and ask the LLM to generate response that are under the budget.
- SEAL [3]: in this method, a steering vector intervention is used during inference to suppress redundant reflection/transition thoughts, thereby shortening reasoning.

In summary, we now provide a baseline collection of six typical SOTA methods across three models and four benchmarks, for a thorough comparison. SEAL does not support Qwen3 model in their official codebase, so we skip the evaluation for SEAL on Qwen3-8B model. The results are summarized in the table below.


|Method|MATH-500_Acc.|MATH-500_#Tokens|AIME-2024_Acc.|AIME-2024_#Tokens|AMC_Acc.|AMC_#Tokens|OlympiadBench_Acc.|OlympiadBench_#Tokens|
|--|--|--|--|--|--|--|--|--|
|*DeepSeek-R1-Distill-Qwen-7B*|||||||||
|Thinking|91.6|2598|36.7|4446|78.3|4338|56.9|3960|
|NoThinking|74.8|-|23.3|-|47.0|-|40.4|-|
|Chain-of-Draft|75.2|690|30.0|2963|50.6|1535|47.7|1911|
|TALE-EP|85.2|1201|30.0|2718|**65.1**|1649|53.9|1693|
|SEAL|87.6|1741|33.3|3932|53.0|2661|52.4|2699|
|Dynasor|83.0|1653|20.0|2743|53.0|2002|48.9|1876|
|Budget Forcing|86.0|1547|16.7|2015|55.4|1872|47.1|1844|
|**Budget Guidance**|**88.2**|1329|**33.3**|2046|60.2|1768|**54.2**|1755|
|*DeepSeek-R1-Distill-Qwen-32B*|||||||||
|Thinking|93.2|2226|70.0|7694|77.1|4156|61.9|3435|
|NoThinking|68.2|-|20.0|-|47.0|-|41.9|-|
|Chain-of-Draft|87.4|918|50.0|2919|68.7|1344|55.4|1283|
|TALE-EP|88.8|1449|50.0|3689|68.7|1879|56.9|1906|
|SEAL|87.2|1495|43.3|3416|55.4|2169|56.3|2220|
|Dynasor|88.0|1978|46.7|4199|67.5|3073|57.5|3079|
|Budget Forcing|86.4|1525|40.0|2936|50.6|1567|50.7|1797|
|**Budget Guidance**|**90.0**|1288|**56.7**|2873|**69.9**|1528|**57.8**|1820|
|*Qwen3-8B*|||||||||
|Thinking|96.2|4613|73.3|13660|91.6|8740|71.7|9424|
|NoThinking|84.8|-|33.3|-|56.6|-|53.5|-|
|Chain-of-Draft|91.6|1891|50.0|3943|78.3|3785|**66.1**|2650|
|TALE-EP|90.8|3006|43.3|5353|79.5|5467|64.2|3604|
|Dynasor|91.2|3062|46.7|5164|72.3|4196|62.7|4064|
|Budget Forcing|90.2|2545|43.3|4010|77.1|3807|61.6|3712|
|**Budget Guidance**|**93.0**|2062|**50.0**|3981|**80.7**|3869|65.6|3639|


For the accuracy comparison, our method generally performs on par with or better than the six baselines across the three models and four benchmarks, while using a similar number of thinking tokens.

We also provide an additional table for a qualitative comparison of these baselines and our method from four aspects in the table below.


| Method| Training-free | Prompt-Free Control | Direct Budget Control | Preserves Full Reasoning Trace |
|--|--|--|--|--|
|NoThinking|✓|✗|✗|✗|
|Chain-of-Draft|✓|✗|✗|✓|
|TALE-EP|✓|✗|✗|✓|
|SEAL|✓|✓|✗|✓|
|Dynasor|✓|✓|✓|✗|
|Budget Forcing|✓|✓|✓|✗|
|**Budget Guidance (ours)**|✓|✓|✓|✓|

As we can see, NoThinking, Chain-of-Draft, and TALE-EP are prompt-based methods and thus heavily rely on rigorous prompt engineering for good performance, and they do not control the reasoning length well according to our experiments. Although SEAL does not rely on prompt engineering and uses a steering vector to guide the reasoning, it does not provide a direct token budget option to control the reasoning under a specific budget. Both Dynasor and Budget Forcing provide direct budget control, but they all rely on some form of early stopping and hard cutoff, and therefore do not preserve the full reasoning trace. Our method is both training-free and prompt-engineering-free, provides direct budget control, and preserves the full reasoning trace by softly steering the entire reasoning process to meet the budget, making it a unique, elegant, and higher-accuracy solution for budget-conditioned generation.



References:

[1] Chain of draft: Thinking faster by writing less

[2] Token-Budget-Aware LLM Reasoning

[3] SEAL: Steerable Reasoning Calibration of Large Language Models for Free

---

> ### Author Response · Authors · 2025-11-26
> **General Response (cont'd)**
>
> > G2. Explanation on why using Gamma distrubition to model the remaining reasoning length.
>
>
> The remaining reasoning length is a strictly positive quantity that can be naturally interpreted as a waiting time until the reasoning process terminates. This aligns with the classical use of Gamma distributions in duration modeling and time-to-event analysis. In contrast, distributions with symmetric or unbounded negative support, like the Gaussian distribution, assign non-zero probability to negative lengths, which is mathematically incompatible with the remaining-length prediction problem. The Gamma distribution, therefore, provides a lightweight and statistically grounded family that matches the structural constraints of the task without imposing assumptions on the underlying generative mechanism of the LLM. The Gamma distribution is not the only option; for example, one can also use nonparametric or mixture models for this purpose. They may offer better modeling but at the cost of more learnable parameters, making training and inference harder. Thus, the Gamma distribution serves as a practical and economical approximation for remaining-length prediction.

---

### Note · Authors · 2026-01-04

I have read and agree with the venue's withdrawal policy on behalf of myself and my co-authors.